# Diet and Proteinuria: State of Art

**DOI:** 10.3390/ijms24010044

**Published:** 2022-12-20

**Authors:** Paolo Ria, Antonio De Pascalis, Anna Zito, Silvia Barbarini, Marcello Napoli, Antonietta Gigante, Gian Pio Sorice

**Affiliations:** 1Nephrology, Dialysis and Renal Transplantation Unit, Vito Fazzi Hospital, 73100 Lecce, Italy; 2Department of Translational and Precision Medicine, Sapienza University of Rome, 00185 Rome, Italy; 3Section of Internal Medicine, Endocrinology, Andrology and Metabolic Diseases, Department of Emergency and Organ Transplantation, University of Bari—Aldo Moro, Piazza Giulio Cesare, 11, 70124 Bari, Italy

**Keywords:** diet, nutrition, proteinuria, low protein

## Abstract

Proteinuria is a broad term used to describe the pathological presence of proteins, including albumin, globulin, Bence-Jones protein, and mucoprotein in the urine. When persistent, proteinuria is a marker of kidney damage and represents a reliable predictor of the risk of progression of renal failure. Medical nutrition therapy is imperative for patients with proteinuria because it may slow the progression of renal disease. The aim of this review is to explore different nutritional approaches in the management of proteinuria and their influence on pathophysiological processes. As such, protein restriction is the main dietary intervention. Indeed, other management approaches are frequently used to reduce it regarding micro and macronutrients, but also the dietary style. Among these, the nutritional approach represents one of the most used and controversial interventions and the studies rarely take the form of randomized and controlled trials. With this work we aspire to analyze current clinical knowledge of how nutrition could influence proteinuria, potentially representing a useful tool in the management of proteinuric nephropathy.

## 1. Introduction

Proteinuria is an independent risk factor for progression to end-stage renal disease. Reducing it, therefore, is an important strategy in delaying or preventing loss of renal function [1]. The pathophysiological causes that correlate proteinuria to kidney damage are various and not entirely known. One of its most important features is the glomerular barrier penetrability alteration, resulting from protease activities and the reduction of the synthesis of proteoglycans [2]. Moreover, reactive oxygen species (ROS) and free radicals play a very important role in the pathogenesis of nephrotic syndromes [3]. Transforming growth factor-beta (TGF-β) cytokine is another factor involved in the process of fibrosis and glomerular sclerosis since it increases the synthesis of the extracellular matrix [4]. Against this background, the scientific community has been searching for answers to the question of how dietary manipulation can affect these pathological changes for several years. Despite the importance of this topic, many studies have so far explored only protein intake. Moreover, the populations included in these studies were often too heterogeneous and the follow-up too short to validate the safety of the treatment, proteinuria values were rarely indicated, and few randomized clinical trials were included.

The aim of this review is to explore different nutritional approaches and their influence on pathophysiological processes. To this end, we investigated the influence of diet and its possible effects on several clinical aspects.

## 2. Low Protein Diet (LPD) and CKD Progression

Glomerular filtration is influenced by dietary protein intake which is responsible for a great fraction of kidney workload. Many experimental and clinical papers have confirmed the renal effects of a protein load and the deleterious role of the renal hyperfiltration response associated with protein intake itself [5]. Therefore, when nephron number is reduced such as in CKD, lowering protein intake will reduce hyperfiltration, with a potential additive effect to those of well-known beneficial drugs such as RAAS inhibitors and recently discovered sodium-glucose transporter 2 inhibitors (SGLT2-i) [6]. As a consequence of reducing uremic toxins on one hand and improving renal hemodynamics, on the other hand, a reduction in protein intake may reduce uremic symptoms and delay the need to start chronic dialysis treatment. As also reported by the KDOQI guidelines, low-protein diets are a means to reduce the progression of chronic nephropathies, in addition to the metabolic control of urea and phosphate levels and acidosis [7].

The effects of a low-protein diet have many similarities with those of SGLT2-i: an increased glomerular filtration of amino acids increases sodium reabsorption in the proximal tubule and with a decreased sodium delivery to the distal nephron; this, in turn, inhibits the tubuloglomerular feedback, lowering the resistance of the afferent arteriole, thus allowing glomerular hyperfiltration, similar to diabetic kidney disease [8]. Furthermore, in CKD patients, a fall of GFR is observed in the first weeks of protein restriction; afterwards, eGFR stabilizes and the disease tends to progress slower than in patients on an unrestricted diet. This observation, well described in the MDRD trial is comparable to the “dip” of eGFR described in the course of treatment with SGLT2-i [9].

Unlike the case of RAAS inhibitors where the synergy between reduced intake of sodium and animal proteins has been largely investigated, to our knowledge no clinical study evaluated the potential beneficial effects of a low protein diet on the action of SGLT2-I [10]. However, a high protein diet induces glomerular hyperfiltration that could blunt the favourable effects of SGLT2-i. In this context restoring a normal protein intake may, at least, allow setting a favourable stage for SGLT2-i activity. 

In addition to the better-described glomerular hemodynamic effect, in animal models, a low-protein diet may reduce tubular cell damage, tubulointerstitial oxidative stress, and apoptosis by decreasing the accumulation of abnormal mitochondria. This finding could be explained by the restoration of autophagy; furthermore, low protein diets reduce the mammalian target of rapamycin complex 1 (mTORC1) activity and promotes autophagic capacity [11].

In conclusion, according to the Nutritional Guidelines in CKD (K-DOQI 2020), in adults with CKD 3–5 (not on dialysis) who are metabolically stable, to reduce the risk for end-stage kidney disease, a low-protein diet providing 0.55–0.60 g dietary protein/kg body weight/day (or a very low-protein diet providing 0.28–0.43 g dietary protein/kg body weight/day with additional keto acid/amino acid) is recommended. In the adult with CKD 3–5 who has diabetes, it is reasonable to prescribe, under close clinical supervision, a dietary protein intake of 0.6–0.8 g/kg body weight per day to maintain a stable nutritional status and optimize glycemic control [7].

## 3. LPD and Proteinuria

A low-protein diet may decrease proteinuria, especially in non-diabetic CKD patients. The decrease in proteinuria is associated with beneficial effects on serum albumin levels and lipid metabolism disorders. In hypoalbuminemic patients, it could promote an increase in serum albumin, obviously fostered by the reduction in urinary protein excretion but also by a complex adaptation of protein metabolism linked to postprandial stimulation of protein synthesis, a decline of whole-body proteolysis, and reduction in amino acid oxidation [12,13]. A plant-based diet may contribute to the control of proteinuria since vegetable proteins have a poorer impact on renal haemodynamics than animal proteins. Switching animal proteins to vegetable proteins may decrease renal hyperfiltration, proteinuria, and, ideally, in the long term, the risk of developing or worsening renal failure [14] (Figure 1). 

The implementation of base-inducing fruits and vegetables into the diet has been demonstrated also to decrease dietary acid load and blood pressure in patients with macroalbuminuric hypertensive nephropathy and stage 2 CKD [15].

In vivo and animal models on this approach have been carried out since the 1980s with Brenner and El-Nahas showing that an LPD reduced hyperfiltration and glomerular sclerosis in rats [16,17]. Since then, many studies have analyzed the effects of LPD, unfortunately, mostly in patients who already had chronic renal failure and rarely in those who as yet had no renal damage. 

The most recent meta-analysis of randomized controlled trials was performed in 2019 and analyzed the effect of a low-protein diet on kidney function (Table 1). It demonstrated that the principal target of an LPD was not the improvement of glomerular filtration rate (GFR), but the reduction in proteinuria. In fact, in 19 studies, LPD had no significant effect on GFR, but inversely had a substantial influence on proteinuria. Dose-response analysis showed that a reduction of protein intake by 0.1 g/kg/d was associated with a 0.0031 g/24 h reduction in proteinuria. Nevertheless, when LPD was continued for over 1 year, proteinuria was significantly reduced (−0.673 g/24 h), and the reduction appeared greater in subjects aged over 60 (−0.526 g/24 h). Furthermore, an LPD was also found to reduce body weight, BMI, urea, and BUN [18].

Chaveau et al., (Table 1) investigated the change in proteinuria in response to a supplemented very low protein diet (sVLPD) in 220 consecutive patients with chronic kidney disease (CKD). The diet provided 0.3 g of vegetable protein plus 1 g of protein per gram of protein >3 g/d. The amount of inorganic phosphorus was 5–7 mg/kg/d and was supplemented with one tablet for every 5 kg body weight of a mixture of essential keto analogs and amino acids. Total energy was 35 kcal/kg/d. The population was divided into 2 groups according to proteinuria at baseline: 1–3 g/d and >3 g/d. In both groups, proteinuria decreased by about 50%, with a greater reduction in patients who had higher baseline proteinuria. The maximum efficacy was achieved after 3 months. The patients with a greater reduction in proteinuria showed a significantly lower decline in GFR. The decrease in proteinuria positively influenced albumin levels and lipid metabolism disorders. The authors hypothesized that short-term antiproteinuric response allowed the prediction of long-term GFR decline and concluded that nutritional therapy should probably be continued in these responder patients [19]. The most important limitations of this study were the small sample size and the fact that all patients had an advanced renal failure with CKD stage IV or V.

## 4. Safety of Protein Restriction

Although many studies have shown the effectiveness of protein restriction in the reduction of proteinuria, several other studies have analyzed the safety of LPD and VLPD. In fact, in this scenario, malnutrition is one of the most serious risks, as LPDs expose the patient to insufficient nutrient intake and protein-energy wasting (PEW). A meta-analysis conducted in 2018 showed that LPD did not cause malnutrition [13] and that it could maintain an adequate nitrogen balance in nephrotic syndrome. Evidence suggests that this balance is maintained because increasing protein loss could promote essential amino acid saving [22]. Various studies have shown the importance of an adequate intake of calories during LPD (30–35 kcal/kg/d) to prevent malnutrition [23]. Furthermore, protein-energy wasting has been associated with VLPD only in the case of insufficient calorie intake [24].

Another study, this time observational, investigated the safety and effectiveness of an LPD supplemented with keto acids in diabetic patients with chronic kidney disease. The results of this study indicated that this diet did not have a negative impact on nutritional markers, body composition, muscle mass, and fitness. In fact, after beginning the LPD diet, muscle strength (measured by a digital hand grip dynamometer) increased in the diabetic group. Moreover, the overall metabolic profile of diabetic patients improved, as did uremia and diabetes. Glucose control and insulin sensitivity improved probably also due to the reduction of urea [25].

In a large meta-analysis of randomized controlled trials, Yue et al. provided a good analysis of the long-term safety of protein restriction by examining LPD lasting longer than one year. Protein restriction significantly reduced BMI (−0.907 kg/m^2^; CI: −1.491 to −0.322 kg/m^2^) and albumin (−1.586 g/L; CI: −5.258 to 2.086 g/L), implying that a long period of treatment may lead to malnutrition. Additionally, LPD reduces the secretion of growth hormone and glucagon. In the MDRD study, post hoc analysis underlined that VLPD increased the long-term risk of mortality in CKD patients [26]. On the other hand, several other studies have not found nutritional deficiency [27] or confirmed a higher risk of malnutrition [13]. In this regard, the use of keto analogues could reduce the risk of malnutrition since supplementation improves nitrogen balance and enhances protein status [28]. Finally, the potential reduction of proteinuria ameliorates serum albumin levels. These effects, however, could be related to other adaptions of protein metabolism, since a series of other mechanisms are activated, namely the reduction of whole-body proteolysis, the reduction in amino acid oxidation, and postprandial stimulation of protein synthesis [20].

Therefore, no definitive conclusions on the safety of LPD and VLPD could be drawn from the above studies as there were several confounding factors, and data on effective protein intake was insufficient.

## 5. Vegetarian Diet, CKD and Proteinuria

Clinical trials on diet and kidney disease usually focus on protein intake but rarely on the type of protein. This difference represents a core point, as highlighted by a Chinese study where higher red meat intake was associated with an increased risk of end-stage renal disease (ESRD) [29].

In 1994 Barsotti et al. demonstrated that the transition from an animal-vegetable diet (1.0–1.3 g/kg/die of protein) to a vegan diet (0.7 g/kg/die) was associated with a significant decrease in proteinuria in patients with non-diabetic nephrosis [30].

A recent cross-sectional study performed in Taipei evaluated the association between vegetarian diets and CKD prevalence. The study enrolled 55,113 individuals, where 4236 were vegans, 11,809 ovo-lacto vegetarians, and 39,068 omnivores. It found that a vegetarian diet was significantly associated with a lower prevalence of CKD. The population studied was heterogeneous, with a 16.8% incidence of CKD and a mean eGFR of 84 mL/min per 1.73 mL/min/m^2^. A lower prevalence of proteinuria was associated with the vegan group. The study, however, had some limitations, among which were a possible selection bias and the lack of information about energy intake and the nutrient composition of the diets [31].

A series of studies have investigated the effectiveness of the use of soy protein in kidney disease. Soy protein is the only high-quality plant-based protein that is widely available and provides essential amino acids. Soy also contains isoflavones, fiber, and oil. In a meta-analysis, soy protein consumption was shown to reduce serum creatinine, serum phosphorus, and triglyceride concentrations compared with animal protein [32]. In an older study, the consumption of at least 25 g/die of soy protein resulted in a decrease in total cholesterol and particularly in LDL cholesterol, in nephrotic patients with proteinuria [33]. There have been conflicting results in terms of proteinuria, as some studies have demonstrated that a soy protein diet reduces proteinuria [34], while other studies have found no differences between soy and meat diets [35]. In contrast to the latter, a study by Anderson et al. found greater urine protein excretion with a soy diet than with an animal protein diet in patients with type two diabetes [36]. Isoflavones could play a key role, as shown by Texeira et al., who found that urinary albumin-creatinine ratios were inversely correlated with plasma isoflavone concentrations [20]. In general, studies have found that soy diets are not dangerous for the kidney and could probably reduce the decline of glomerular filtrate and progression of proteinuria in the long term. 

## 6. LPD and Keto Analogues

A small open-label, randomized, controlled, single-center clinical trial on 17 patients with hepatitis B virus infection and chronic glomerulonephritis, evaluated the effects of low-protein intake on proteinuria and nutritional status (Table 1). All the patients had CKD stage I–II and proteinuria > 1 g/d. For one year, 9 patients followed an LPD regimen (0.6–0.8 g/kg/d of ideal body weight) without supplementation while 8 patients received LPD with a keto acid supplementation (0.1 g/kg/d). In the supplemented-LPD group, 24-h total proteinuria values were significantly lower than baseline after six months and one year. Moreover, proteinuria was significantly lower in the group with keto analogues than in the group without supplementation (2.0 ± 1.8 vs. 4.4 ± 2.7 g/24 h). Lastly, during the treatment, the group without keto analogues showed no change in proteinuria [37]. This study demonstrated that LPD supplemented with keto acids significantly improved proteinuria and malnutrition, compared with a non-supplemented diet. However, the study was limited to only a few patients with a specific pathology. This small trial supported the theory that keto acids decrease levels of fibrotic factors such as TGF-β. Furthermore, it appears that keto analogues improve oxidative stress injury induced by an LPD in nephrectomized rats [21]. Moreover, keto acid treatment could directly reduce iPTH secretion and serum phosphate [38], improving the calcium-phosphorus balance and reducing the risk of adynamic bone disease. 

These studies show that keto analogues are not only indicated to reduce the risk of malnutrition but have a “direct” role in decreasing proteinuria and in nephroprotection.

## 7. Diabetes Mellitus

Several clinical investigations and meta-analyses have shown that protein restriction delays the initiation of renal replacement therapy in patients with CKD in type two diabetes; this delay is linked to a reduction in the rate of decline of the GFR [39]. A low-protein diet appears to be able to reduce intraglomerular pressure and glomerular plasma flow. It has been hypothesized that the haemodynamic effects of a hypo-proteic diet are due to modifications in pre-glomerular and/or post-glomerular vascular resistance and the tubuloglomerular feedback system. The non-haemodynamic effects of a hypo-proteic diet (reduction of plasma protein uptake in the mesangium, decrease of renal expression of growth factors) are well known [11]. A low-protein diet seems to be effective in reducing the progression of renal damage even in diabetic patients, although data are not conclusive. A recent meta-analysis published in 2019 analyzed the efficacy of LPD in diabetic nephropathy (Table 1). The data were comparable with the results observed in non-diabetic subjects. There were no significant differences in serum creatinine, glomerular filtration rate, and glycosylated hemoglobin. Urinary albumin excretion rate and proteinuria were significantly lower in the LPD group versus the control group (standard mean difference: 0.62, 95% CI 0.06–1.19 and 0.69, 95% CI 0.22–1.16 respectively) [40]. A systematic review by Zhu et al. also confirmed these results by finding significantly lower proteinuria in the LPD patients in the type two diabetes subgroups (1.32, 95% CI 0.17–2.47, *p* = 0.02) [41]. The authors tried to address the nephroprotection provided by LPD from a pathophysiological perspective. In particular, the reduced protein load implied: first, the inhibition of the intrarenal renin-angiotensin system (RAS); second, the reduction of glucagon secretion with a lesser dilatation of afferent arterioles in the glomeruli; third, the reduction of insulin-like growth factor-1 secretion, which has a powerful vasodilator action [42]. Furthermore, LPD reactivates autophagy through the suppression of the mechanistic target of the rapamycin complex 1 (mTORC1) pathway in the type two diabetes animal model [43]. In general, these two studies reported a decrease in proteinuria but a modest efficacy of LPD as a nephroprotective diet in diabetic nephropathy. A recent meta-analysis, performed on studies conducted on patients with type one diabetes and established microalbuminuria or nephropathy, has provided positive results, although not all these studies are methodologically rigorous [44]. Recently, moderate dietary protein restriction has been shown to improve prognosis in type one diabetic patients with CKD, although it did not significantly modify the decline in GFR [45]. On the contrary, as mentioned, the MDRD study, in which, however, only 3% of patients had type two diabetes (and none had type one diabetes), showed no evident benefits of protein restriction on kidney disease progression over a period of 2–3 years.

In an innovative crossover-controlled trial in seventeen type two diabetes patients, De Mello et al. analyzed the differences between a usual diet, a chicken- diet (without other types of meat), and a lactovegetarian LPD regimen. The urinary albumin excretion rate was significantly lower after the chicken diet (CD) and lactovegetarian diet compared to the usual diet (20.6%, 95% CI 4.8–36.4% and 31.4%, 95% CI 12.7–50% respectively). The reduction in albuminuria after the chicken and LPD did not show a significant difference (*p* = 0.249) [46]. Reasonably, the CD and LPD increased serum polyunsaturated fatty acids (PUFAs) that influenced the decrease in proteinuria. Another hypothesis considered the increase in PUFAs as related to the decrease of proteinuria [47]. High levels of PUFAs could have a positive effect on endothelial function and could improve insulin resistance with a favorable effect on proteinuria. However, the study was limited due to the small number of patients. 

Finally, in the presence of a pharmacological blockade of the RAS, protein restriction does not seem to produce any additional effects [48]. Based on these elements, it is believed that patients with overt nephropathy must maintain a protein intake of 0.8 g/kg/day (approximately 10% of the total daily calorie intake [49].

As eGFR begins to decline, a further reduction to 0.6 g/kg/day may be effective in slowing its progressive reduction in selected subjects. However, diabetic patients have greater difficulties in tolerating a severe restriction in dietary proteins in addition to the usual dietary regime and are also exposed to a higher risk of malnutrition than non-diabetic subjects, determined by the increased endogenous catabolism induced by insulin deficiency. Consequently, dietary intervention should be evaluated by a dietician and diabetologist with considerable experience and the patient's nutritional status must be carefully monitored, in order to avoid the onset of malnutrition, which is an unfavorable prognostic factor [50,51].

In addition to the possible benefits on the progression of renal failure, protein restriction (and the associated reduction in the content of phosphorus and saturated fat) is useful for correcting, at least in part, the metabolic alterations that characterize this condition (hyperparathyroidism, dyslipidemia, hyperazotemia, and metabolic acidosis).

## 8. New Approaches

Apart from protein intake, an interesting study has shown that curcumin attenuates albumin urinary excretion in type two diabetic patients [52]. In the study, 14 patients were treated with an oral dose of 500 mg per day of curcumin for 15 days. After the 15-day curcumin treatment, the amount of urinary albumin decreased by 70% (*p* < 0.05). This effect was maintained after 30 days. Efficacy was greater in patients with diabetic kidney disease. Based on this study, it has been argued that hyperglycemia-induced oxidative stress also alters the nuclear factor erythroid-derived-2-like 2 (Nrf2) anti-oxidative system. Various animal studies have demonstrated that curcumin was an effective Nrf2-activator [53], prevented β-cell death, and attenuated insulin resistance [54]. Other important effects of curcumin were a reduction of malondialdehyde (a lipid oxidation index), a 25% decrease in inflammation by reducing levels of lipopolysaccharide through modulation of gut microbiota (after curcumin treatment, different bacteria growth significantly contributes to maintaining a balance of gut microbiota and gut barrier), and inhibition of apoptotic signaling proteins in blood lymphocytes (Table 2).

## 9. Dysbiosis and the Immune System

Based on current knowledge it is possible to speculate that diet could cause dysbiosis and alteration of microbiota composition, which could lead to aberrant immune responses and increased risk of disease [69]. As previously reported, gut microbiota plays a fundamental role in the development of local and systemic immune systems. In adults, the two principal bacterial phyla are Bacteroides and Firmicutes [70]. Bifidobacteria spp promote the maturation of the mucosal IgA system. During adulthood, the microbiota drives a regulatory mechanism intended to maintain both mucosal and systemic immunity. Many species of bacteria can influence immune response. Polysaccharide A associated with B. Fragilis promotes Th1/Th2 balance [71], while Filamentous bacteria induce Th17 responses and increase the number of T-regulatory (Treg) cells [72]. Complex carbohydrates increase levels of beneficial Bifidobacteria spp [73]. High-fat diets enriched with omega-6 polyunsaturated fatty acid (PUFA) increase the population of Firmicutes and decrease Bacteroidetes in rats [74]. In vegetarian diets, the higher fiber intake promotes the production of short-chain fatty acids by microbes, which decreases intestinal pH. This acidification reduces the growth of potentially pathogenic bacteria [75]. Studies have found that these changes in natural homeostasis are associated with different pathologies. For example, the increase in Bacteroides/Firmicutes ratio has been associated with insulin resistance or type two diabetes [76]. It is thus intuitive that prebiotics and probiotics can play an important role in the reduction of inflammatory and immunological diseases.

In this context, an interesting experimental study highlighted how gut *Lactobacillus* protects against the progression of renal damage by modulating the gut environment in rats [57]. In this study, *Lactobacillus* supplementation decreased proteinuria in rats with CKD by restoring the expression of intestinal barrier proteins and reducing systemic inflammation. In uremic patients, the microbiota was altered and probiotics, in particular *L. Acidophilus*, *Bifidobacterium longum*, and *Streptococcus thermophiles* could have reduced BUN [77]. It should also be noted, however, that many studies have reported no beneficial effects of the use of probiotics in renal damage [78].

## 10. Dietary Style, Macro- and Micro-Nutrients

Medical nutrition therapy is imperative for patients with proteinuria because it may slow the progression of renal disease. It is important to reiterate that nutritional intervention concerns both macronutrients and micronutrients. In the previous sections, the influence of the quantity and type of protein intake was mainly analysed. It was also analyzed how the carbohydrate intake must be adequate to avoid states of malnutrition but also to avoid the onset of obesity, metabolic syndrome, and diabetes which represent further renal and cardiovascular risk factors. Even the fat intake must be rationalized, increasing the monounsaturated and polyunsaturated ones and limiting processed oil and saturated fat. The most current diet recommendations for cardiovascular prevention (such as food plant-based diet, Mediterranean Diet, or DASH diet), have also been effective in preventing and delaying the progression of proteinuria and renal disease in general [79,80]. They usually have a relatively low protein intake, especially of animal origin, are rich in polyunsaturated fatty acids, are rich in fiber, and have an adequate percentage of carbohydrates. Furthermore, they have appropriate sodium and phosphorus levels. Lastly, the DASH diet may be protective against the progression of CKD and is high in potassium and low in sodium [81].

Moreover, an unconventional study analysed how not only the type of diet can influence kidney disease but also the frequency of meals. This study showed how skipping breakfast and dinner were identified as risk factors for proteinuria in females, but not in males [82]. Another cross-sectional study, with 60,800 Japanese adults, reported a significant association between skipping breakfast and the high prevalence of proteinuria in both males and females [83]. A randomized controlled study highlighted how skipping breakfast was associated with postprandial hyperglycemia after subsequent lunch and even dinner [84]. This condition induces oxidative stress and vascular endothelial dysfunction [85]. Furthermore, late dinner was associated with a higher risk of proteinuria onset. Currently, the pathogenetic mechanisms underlying this correlation are not fully understood [86].

Recent studies show that micronutrients are also important elements in the nutritional therapy of proteinuria. In the next part of this section, we analyze how some of these elements are involved in the management of proteinuria.

### 10.1. Alkali, and Vitamin K

The management of proteinuria through new foods, such as curcumin, has already been analyzed. Other nutrients that could interact with renal outcomes have also been studied: i.e., fiber, alkali, and vitamin K1. Fiber and vitamin K1 content was found to be higher in the vegan diet and VLPD than in other diets. These nutrients increased the alkalizing potential [56], which could improve the effectiveness of such diets, beyond the benefits derived from the reduction of protein intake. Vitamin K has been associated with a reduction in mortality in people with chronic kidney disease [57] and, as mentioned previously, fiber intake was seen to decrease intestinal pH and favorably modulate microbiota. Furthermore, reducing dietary acid load could reduce mortality in people with chronic kidney disease, as it can improve the acid-bases homeostasis and optimize the control of hyperkalemia, especially combined with treatment with angiotensin-converting enzyme (ACE) inhibition and/or angiotensin receptor blocker [58].

### 10.2. Phosphorus

Serum phosphorus and phosphate intake are also important in the management of proteinuria. Lee et al. demonstrated that higher serum phosphorus, even in the non-pathological range, was independently and positively related to low-grade albuminuria and was a potent predictor of an increase in urinary albumin-to-creatinine ratios (regression coefficient = 0.610, *p* < 0.001). This study did not include patients with eGFR < 60 mL/min and with proteinuria or microhematuria [59]. Dietary phosphorus intake, particularly with animal proteins, has been shown to increase serum phosphorus levels and decrease flow-mediated dilatation, a substitute marker of endothelial function [60]. Moreover, another study in CKD patients confirmed that phosphate attenuated the anti-proteinuric effect of VLPD [61]. Lastly, high levels of phosphate attenuated the nephroprotective effect of ACE-inhibition in patients with proteinuria and CKD [62].

### 10.3. Sodium Intake

Sodium intake is crucial in the nutritional approach to proteinuria. An interesting randomized trial showed that the reduction of proteinuria, through moderate dietary sodium restriction added to angiotensin-converting enzyme (ACE) inhibition was significantly higher compared to the effect obtained by the addition of angiotensin receptor blockade to ACE inhibition. In patients undergoing ACE inhibition with an ad libitum diet, proteinuria was 1.68 g/d (1.31–2.14). The addition of angiotensin receptor blockade decreased proteinuria to 1.44 g/d (1.07–1.93; *p* = 0.003), while the addition of a low-sodium diet to ACE-inhibition reduced proteinuria to 0.85 g/d (0.66–1.10; *p* < 0.001). Proteinuria was the lowest (0.67 g/d, 0.50–0.91; *p* < 0.001) in patients with dual blockade plus sodium restriction [63]. None of the patients studied had diabetic nephropathy. Furthermore, other studies have reported an additive antiproteinuric effect in patients with ACE-inhibition and LPD, and VLPD. The reciprocally additional effect was explained by two different mechanisms: glomerular preload, and afterload reduction [64]. Typically, LPD has a lower intake of sodium, but in general, sodium- and protein-restriction, associated with drugs that inhibit RAS is an effective strategy to reduce proteinuria (Table 2).

### 10.4. Potassium Intake

Potassium intake affects kidney function, but also kidney function affects potassium handling. We have already extensively discussed the relationship between a low-sodium diet and its effects on proteinuria. It is intuitive to consider that individuals with a DASH diet (rich in fruit and vegetables) would have a lower salt and high potassium-containing diet and therefore have lower blood pressure which would reduce the degree of proteinuria. An interesting meta-analysis of the MDRD cohort showed lower blood pressure with potassium supplementation, in hypertensive individuals, especially when they consume high sodium and low potassium and are not taking antihypertensive medication [87]. Santhanam et al. conducted a study, on the MDRD cohort, that looked at the association between dietary potassium intake, BMI, and proteinuria in normotensive and hypertensive individuals, and reported a negative correlation between dietary potassium and proteinuria in normotensive individuals [88]. These studies confirm what has been demonstrated in recent years regarding plasma potassium modulating sodium reabsorption through the sodium chloride cotransporter (NCC). Higher potassium inhibits sodium reclamation via NCC, with subsequent BP reduction, vascular resistance reduction, and kidney protection. Diets high in potassium intake may also directly improve kidney function through increased kallikrein expression which has been demonstrated to reduce kidney injury in animal models [65]. Smyth et al. performed a post hoc analysis of the ONGOING and TRASCEND cohort trials and found that higher 24-h urinary K+ excretion was associated with a lower risk of eGFR decline, progression of proteinuria, and initiation of dialysis [66]. Of interest, subgroup analysis showed a loss of this association in individuals with more severe CKD (eGFR <45 mL/min/1.73 m^2^).

Ellis et al. also reported a reduction in albuminuria and improvement in hypertensive kidney lesions in K+ supplemented, salt-loaded spontaneously hypertensive rats. High dietary K+ intake resulted in less collagen Types I and III depositions in the kidney, macrophage infiltration, and lower expression of the inflammatory cytokines [67].

Sodium, acid-base balance, and potassium are intimately linked; furthermore, a diet rich in potassium, as described, or potassium citrate o potassium bicarbonate supplements, reduces urinary protein excretion, acting on sodium reabsorption and so improves blood pressure and glomerular filtration. Additionally, it improves metabolic acidosis in patients with CKD, especially in the early stages [68].

Further clinical trials are needed to determine the optimal dietary potassium intake to achieve a benefit, identifying the level of plasma potassium at every stage of CKD, that is detrimental, and suggesting approaches to achieve these goals.

## 11. Conclusions

The aim of this work was to provide a review of the nutritional approach to proteinuria. As already mentioned, there are few studies on the effect of diet on proteinuria, and they are often post hoc analyses and frequently limited to patients with CKD. Moreover, they rarely report baseline proteinuria values. Despite this, the various studies lead to endorsable conclusions. In particular, although we certainly cannot assume that the various nutritional approaches influence eGFR, they are effective on proteinuria, which remains a pathological situation that should not be underestimated. 

Regarding the type of proteins, a vegetarian diet is still optimal for patients with chronic disease and with other comorbidities, even if the results are not always encouraging. A systematic review has reported that vegetarian diets are appropriate for individuals during all stages of the lifecycle, including pregnancy, lactation, infancy, and childhood [89]. While it is reasonable to assume that vegetable proteins are preferable to animal proteins in the management of proteinuria, especially in chronic kidney disease, the results of the study are, however, not conclusive. The guidelines relating to protein intake recommend that it be slightly increased in patients with nephrotic syndrome. However, an individualized approach with a hypo-normoprotein diet, with adequate caloric intake and eventually supplementation with keto analogs, could be a reasonable therapeutic option.

Healthy nutrition could have a positive effect on patients with proteinuria and reduce its onset. Nutritional status should be continuously monitored, especially in patients on a hypoproteic diet. Moreover, the improvement of the microbiota is one of the most important pathophysiological causes of proteinuria reduction.

Finally, guidelines in proteinuria management are lacking; nevertheless, each patient should have personalized nutritional therapy, based on the values and causes of proteinuria, the comorbidities, and the nutritional status. Only a tailored approach can increase compliance and outcomes while reducing adverse effects.

## Figures and Tables

**Figure 1 ijms-24-00044-f001:**
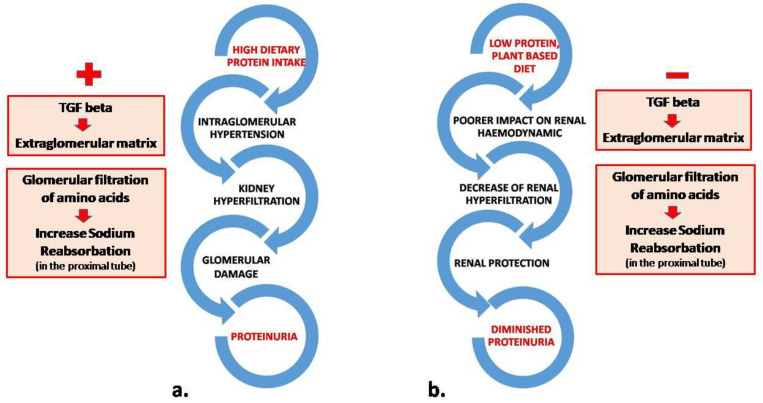
Mechanistic model associating dietary interventions and reduction of proteinuria. High dietary protein intake (**a**) leads to intraglomerular hypertension eliciting high levels of TGF-beta and subsequent accumulation of extraglomerular matrix with the development of fibrosis. During a low protein diet (**b**), renal hemodynamics is preserved and fibrosis TGF beta-mediated is avoided.

**Table 1 ijms-24-00044-t001:** Principal clinical studies of protein intake in proteinuria.

Study	Patients	Renal Function	Protein Intake (g/kg/d)	Clinical Information	Results
Yue, H. et al., 2019 [18] Metanalysis	3566	NA	0.28–0.8 g/kg/d	NA	When LPD > 1-year reduction of protein intake by 0.1 g/kg/d was associated with a—0.673 g/24 h reduction in proteinuria.
Chauveau, P. et. al., 2007 [19]	220	CKD IV-V	0.3 (vegetarian) + 1 g per gram of protein > 3 g/d + supplement	NA	Proteinuria reduction 50%. Max efficacy after 3 months. Greater reduction of proteinuria = lower decline in GFR.
Li, H. et. al., 2019 [20] Metanalysis	690	NA	0.6–1.0 g/kg/d	Diabetes	Proteinuria decreased in the LPD group vs. control group (SMD respectively: 0.62, CI 0.06–1.19 and 0.69, CI 0.22–1.16)
Mou, S. et. al., 2013 [21]	17	CKD I-II and proteinuria > 1 g/d	0.6–0.8 g/kg/d of ideal body weight with supplement (0.1 g/kg/d) or without	HBV+	Proteinuria was significantly lower in the group with keto analogues than in the group without supplementation (2.0 ± 1.8 vs. 4.4 ± 2.7 g/24 h).

**Table 2 ijms-24-00044-t002:** Mechanism of action and results of different nutrients in proteinuria management.

Nutrients	Mechanism of Action	Results
Curcumin (Clinical and animal model) [52,53,54]	Nrf2-activator; prevents β-cell death; attenuates insulin resistance; reduces inflammation	Attenuated urinary excretion of albumin in type two diabetic patients
*Lactobacillus* (Animal model) [55]	Modulates gut environment; restores expression of intestinal barrier proteins; reduces systemic inflammation	Decreased proteinuria in rats with CKD
Alkali and Vitamin K (Clinical Study) [56,57,58]	Decreases intestinal pH and favourably modulates microbiota	Reduce dietary acid load; reduced mortality in people with chronic kidney disease; improved acid-bases homeostasis; improved control of hyperkalaemia
Phosphorus (Clinical Study) [59,60,61,62]	Decreases endothelial flow-mediated dilation; attenuates the anti-proteinuric effect of VLPD and ACE-inhibition	Independently and positively related to low-grade albuminuria; increased urinary albumin-to-creatinine ratios
Sodium intake reduction (Clinical Study) [63,64]	Reduces glomerular preload; inhibition of RAS	Reduction of proteinuria; cumulative antiproteinuric effect when associated with ACE inhibitor and angiotensin receptor blockade
Potassium (Clinical and animal model) [65,66,67,68]	Modulates sodium reabsorption through the sodium chloride cotransporter (inhibits sodium recovery). Reduce the expression of inflammatory cytokines. Increases Kallikrein	Reduced urinary protein excretion, improves blood pressure, glomerular filtration, and improves metabolic acidosis in patients with CKD

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
