# Peer review of "Diet and Proteinuria: State of Art"

_ijms, 2022, doi:10.3390/ijms24010044_

Round 1

Reviewer 1 Report

The review is interesting and deals with a little explored topic, but very important. The text is comprehensive by exploring the various disease options most commonly as causes of chronic kidney disease. The references are in accordance with the above-mentioned observations. In cconclusion I suggest that it be accepted for publication.

Reviewer 2 Report

The present manuscript (Manuscript ID: ijms- 2006292) discusses potential dietary manipulations in the management of proteinuria, suggesting these manipulations as an important target for clinical intervention. The review is interesting, however, there are some points that need to be addressed.

Comments to be addressed:

1)    The manuscript should be updated and include studies of recent literature.

2)    Please edit the abstract in order to be more informative.

3)    Please discuss the impact of potassium salts manipulation on proteinuria.

4)    Please expand on the mechanistic part that associates dietary interventions and reduction of proteinuria. A schematic   model would be useful.

Round 2

Reviewer 2 Report

The authors of the current manuscript (ijms-2006292) have incorporated the required changes in the revised version. However, a minor comment still needs to be addressed.

Minor comment

In Figure 1, please:

a) Label each model/set of cartoons (e.g. A and B).

b) Each cartoon needs to be informative by providing molecular mechanisms/changes that is associated with.

c) Figure legend description is missing. Please add.

Author Response

We thank the reviewer for giving us the opportunity to better clarify the Figure, now changed accordinlgy.
